# A Novel Plasma-Based Methylation Panel for Upper Gastrointestinal Cancer Early Detection

**DOI:** 10.3390/cancers14215282

**Published:** 2022-10-27

**Authors:** Cheng Peng, Guodong Zhao, Bing Pei, Kai Wang, Hui Li, Sujuan Fei, Lishuang Song, Chunkai Wang, Shangmin Xiong, Ying Xue, Qibin He, Minxue Zheng

**Affiliations:** 1Department of Gastroenterology, The Affiliated Jiangning Hospital of Nanjing Medical University, Nanjing 211100, China; 2Zhejiang University Kunshan Biotechnology Laboratory, Zhejiang University Kunshan Innovation Institute, Kunshan 215300, China; 3Department of R&D, Suzhou VersaBio Technologies Co. Ltd., Kunshan 215300, China; 4Department of Clinical Laboratory, The Affiliated Suqian First People’s Hospital of Nanjing Medical University, Suqian 223800, China; 5Department of Gastroenterology, Affiliated Hospital of Xuzhou Medical University, Xuzhou 221002, China; 6Department of Gastroenterology, First People’s Hospital of Xuzhou, Xuzhou 221002, China; 7Center for Reproduction and Genetics, The Affiliated Suzhou Hospital of Nanjing Medical University, Suzhou 215000, China; 8Suzhou Institute of Biomedical Engineering and Technology, Chinese Academy of Sciences, Suzhou 215163, China

**Keywords:** upper gastrointestinal cancer, DNA methylation, plasma, panel, early detection

## Abstract

**Simple Summary:**

Upper gastrointestinal cancer is a major cancer type in China with low 5-year survival rates due to without cost-effective non-invasive early detection tool. In this study, a novel non-invasive panel was developed for early detection of upper gastrointestinal cancer, and the selected methylated makers in the panel showed excellent PCR amplification efficiency and reproducibility. The panel detected three types of upper gastrointestinal cancers with relative high sensitivity and specificity. We hope this novel tool can help Chinese population to increase the proportion of early diagnosis and treatment of upper gastrointestinal cancer and reduce its incidence and mortality.

**Abstract:**

Background: Upper gastrointestinal cancer (UGC) is an important cause of cancer death in China, with low five-year survival rates due to the majority of UGC patients being diagnosed at an advanced stage. Therefore, there is an urgent need to develop cost-effective, reliable and non-invasive methods for the early detection of UGC. Methods: A novel plasma-based methylation panel combining simultaneous detection of three methylated biomarkers (*ELMO1*, *ZNF582* and *TFPI2*) and an internal control gene were developed and used to examine plasma samples from 186 UGC patients and 190 control subjects. Results: The results indicated excellent PCR amplification efficiency and reproducibility of *ELMO1*, *ZNF582* and *TFPI2* in the range of 10–100,000 copies per PCR reaction of fully methylated genomic DNA. The methylation levels of *ELMO1*, *ZNF582* and *TFPI2* were significantly higher in UGC samples than those in control subjects. The sensitivities of *ELMO1*, *ZNF582* and *TFPI2* alone for UGC detection were 32.3%, 61.3% and 30.6%, respectively; when three markers were combined, the sensitivity was improved to 71.0%, with a specificity of 90.0%, and the area under the curve (AUC) was 0.870 (95% CI: 0.832–0.902). Conclusion: Methylated *ELMO1*, *ZNF582* and *TFPI2* were specific for UGC and the three-methylated gene panel provided an alternative non-invasive choice for UGC early detection.

## 1. Introduction

Upper gastrointestinal cancer (UGC), including gastric cancer (GC), esophagogastric junction cancer (EJC) and esophageal cancer (EC), represent more than 8.7% (1,693,203) of new cancer cases and about 13.2% (1,312,869) of cancer-related deaths worldwide in 2020 [1]. China is the country that suffers the heaviest disease burden of UGC, with more than 40% of new cases and deaths of GC in the world found in China ever year [2], and almost half of the new EC cases all over the world occurring in China [3]. Due to the fact that most UGCs were diagnosed and treated at an advanced stage, the average five-year survival rate for UGC is only about 10–30% [2,4]. However, if the UGCs were detected at an early stage, the five-year survival rate can be as high as 90% [2] and most of them can be curatively treated by endoscopy [5].

Routine screening and early detection of UGCs have proved to be an effective strategy to reduce their incidence and mortality. Although the incidence rate of GC in Japan and South Korea was significantly higher than that in China, the mortality rate of GC was reduced by approximately 67%, due to the high participation rate in screening [6]. The gold standard for the detection and treatment is endoscopy; however, the limitation of medical resources and endoscopists, and low compliance with endoscopy made it difficult for endoscopy to be a primary screening and early detection method for UGC. Although several blood-based tumor biomarkers (such as CEA, CA199, CA724 and SCC-Ag) have been applied for detecting GC or EC, low sensitivities and specificities limited their application [7], and none of them can be simultaneously applied to detect multiple types of UGC. Therefore, there is an urgent need to develop cost-effective, reliable and non-invasive methods for the early detection of UGC.

Over the past decade, the fast development of liquid biopsy made cell-free nucleic acids, such as miRNA, LncRNA or ctDNA, for early cancer detection and the monitoring of therapeutic efficacy from study moving to clinical application [8], especially for miRNA and DNA methylation. Jimmy et al. reported a 12-miRNA panel for GC screening, with a sensitivity of 87% at a specificity of 68% [9]; and Jinsei et al. developed an 8-miRNA panel for the early detection of esophageal squamous cell carcinoma, with sensitivities ranging from 87–89% and specificities ranging from 60–85% [4]. DNA methylation is one of most common epigenetic modifications in mammals, which plays a key role in cancer development; thus, DNA methylation may serve as a biomarker for cancer detection or prognosis [10]. Compared with miRNA, the DNA methylation biomarker is more stable and specific for each cancer type, and the whole process procedure for the DNA methylation test is simpler than that of the miRNA-based test; thus, DNA methylation is more suitable as a biomarker for the early detection of cancer than other biomarkers [11]. Up to now, various DNA methylation-based biomarkers for the early detection of gastrointestinal cancer have been reported [12,13,14], and several commercial kits for colorectal cancer (CRC) screening or early detection have been approved by the FDA and NMPA [15,16,17]. The successful application of DNA methylation markers in CRC early detection indicated that they may serve as a potential effective method for UGC detection. *ELMO1* has been used as a methylated marker for GC detection in plasma [18]; methylated *ZNF582*and methylated *TFPI2* were reported as potential makers for EC [19,20]. However, the sensitivities for those genes detected GC or EC were insufficient when used alone; the experience in our research group indicated that the combination of multiplex methylation markers could significantly improve the sensitivity for early-stage cancer [13], and these three methylation markers have never been used in combination for UGC early detection. Therefore, in this study, we developed a novel plasma-based panel, including these three methylation markers, to examine its feasibility and performance for the early detection of UGC.

## 2. Materials and Methods

### 2.1. Sample Collection

In this case-control study, the performance of a plasma-based multiplex DNA methylation assay designed for UGC detection was evaluated by multi-center cohorts. Three hundred and seventy-six plasma samples were collected from The Affiliated Jiangning Hospital of Nanjing Medical University, The Affiliated Suqian First People’s Hospital of Nanjing Medical University, Affiliated Hospital of Xuzhou Medical University and First People’s Hospital of Xuzhou from July 1, 2020 to June 30, 2022. Including 109 GC patients, 29 plasma samples for EGC, 48 EC patients and 190 control subjects, all the participants were confirmed by endoscopy and patients were examined by pathological diagnoses. The control group contained non-UGC patients, such as non-atrophic gastritis, esophagitis, superficial gastritis and subjects with no evidence of disease.

The inclusion criteria for all the participants were as following: aged at 18 years old or above, no history gastrointestinal cancers, no pregnancy, underwent complete endoscopy, and the patients’ results were confirmed by pathology. A blood sample of 10 mL was collected from each subject using a 10 mL K2EDTA tube and stored at room temperature (20 ± 5 °C) for no more than 4 h. Then, the plasma was separated after centrifugation (twice) and immediately stored at −80°C for long-term storage. This study was approved by the Institutional Review Boards of the Affiliated Jiangning Hospital of Nanjing Medical University (Ethics Committee reference number: 20190438). The informed consent was signed by each participant prior to the sample collection.

### 2.2. Cell Free DNA Extraction, Bisulfite Treatment 

Cell-free DNA (cfDNA) was isolated from 3.5 mL of plasma using the Versa-Autopure nucleic acid purification system (Suzhou VersaBio Technologies Co., Ltd., Kunshan, China). After the lysis and washing steps, the samples were finally eluted in 100 μL of elution buffer. Purified DNA was bisulfite-treated using a fast bisulfite conversion kit (Suzhou VersaBio Technologies Co., Ltd.) according to our previous study [7]. Purification of the converted products was conducted using the Versa-Autopure nucleic acid purification system by three washing steps, followed by the final elution in 100 μL of elution buffer.

### 2.3. Quantitative Methylation-Specific PCR

The plasma cfDNA was analyzed by a multiplex quantitative methylation-specific PCR (qMSP) panel obtained from Suzhou VersaBio Technologies Co., Ltd. (Kunshan, China). This one-tube multiplex panel included the detection and analysis of an internal control (*ACTB*) and 3 methylated genes: *ELMO1*, *ZNF582* and *TFPI2*. Primers and probes included in the panel were presented in Appendix A. qMSP, with a total reaction volume of 30 µL, including 15 µL of cfDNA performed on an ABI 7500 instrument (Applied Biosystems, Foster City, CA, USA). The qMSP reaction conditions were as follows: stage I, 20 min of initial activation at 95 °C; stage II, 50 cycles at 95 °C for 10 s, 58 °C for 30 s and 72 °C for 15 s; and stage III, cooling to 40 °C for 30 s.

### 2.4. Analytic Performance Analysis

To determine the reproducibility and amplification efficiency of each methylated marker in the panel, fully methylated genomic DNA (HCT116 cell line) was diluted to create a series of mixtures (100,000, 10,000, 1000, 100 and 10 copies per reaction), and PCR reactions at each concentration were repeated 3 times. The mean Ct value at each concentration was used to calculate the PCR amplification efficiency of each methylated marker following the formula below:E = (10^−1/slope^ − 1) × 100%

### 2.5. Data Analysis

The result was considered ‘valid’ if the *ACTB* Ct value was no more than 35.0. The cut-off Ct values for *ELMO1*, *ZNF582* and *TFPI2* were 42.0, 35.0 and 45.0, respectively. A receiver operating characteristic (ROC) curve was plotted according to the Ct values and the values of the area under the curve (AUC) were calculated. The Ct values were set to the maximal PCR cycle numbers of 50 for those subjects with no amplification signals in the qMSP reaction [6]. A multinomial logistics regression was applied to obtain the probability, which was used as the test variable to run a ROC curve for the three biomarkers combined. GraphPad Prism 6.0 was used for all the statistical analysis; the Pearson chi-squared test at a significance level of *p* < 0.05 was used for the sensitivity comparison among groups; and the Mann–Whitney U tested for the differences in methylation levels.

## 3. Results

This case-control study included 186 patients with UGC; of these patients, 136 were males and 50 were females, and their median age was 68 years old. For the details, 48 EC patients, 29 EJC patients and 109 GC patients were included, and the percentage of male patients was 87.5%, 75.9% and 66.1%, respectively. The median ages (range) for EC, EJC and GC patients were 68 (52–90), 71 (50–87) and 64 (28–86), respectively. The control group included 74 males and 116 females, where the median age was 45 years old (Table 1).

Before evaluating the performance of the panel in the plasma sample, we verified its reproducibility and amplification efficiency in simulated samples to confirm the effectiveness of the panel. As shown in Figure 1, the reproducibility of *ELMO1*, *ZNF582* and *TFPI2* in the range of 10–100,000 copies/PCR reaction of fully methylated genomic DNA were excellent, and the calibration curves plot by Ct values indicated the excellent linearity (R^2^ ≥ 0.9999) and good PCR amplification efficiency (90% < E < 110%). Therefore, the panel developed in this study could efficiently detect methylated cfDNA fragments as few as 10 copies per reaction.

The DNA methylation levels of *ELMO1*, *ZNF582* and *TFPI2* were analyzed by the mean Ct values from different samples. As shown in Figure 2A and 2C, *ELMO1* and *TFPI2* in the GC, EJC and EC samples all displayed significantly higher methylation levels than those in the control subjects (*p* < 0.01), and the methylation among GC, EJC and EC patients showed no significant difference. While *ZNF582* in the GC, EJC and EC samples showed significantly higher methylation levels than that in the control subjects (*p* < 0.01), the methylation levels in EC were also significantly higher than those in GC (*p* < 0.01) (Figure 2B).

For evaluating the feasibility and performance of the methylation panel for detecting UGC in plasma, we analyzed the sensitivity, specificity and Youden index of each of the methylation markers in different cancer types. As shown in Figure 3A, the sensitivities of *ELMO1*, *ZNF582* and *TFPI2* alone for detecting GC were 33.9%, 56.0% and 27.5%, respectively; the combined used of *ELMO1* and *ZNF582* improved the sensitivity to 67.0% and the combination of three methylation markers could further improve the sensitivity to 67.9%. In the EJC samples, *ZNF582* also showed a relatively higher sensitivity than *ELMO1* and *TFPI2* (62.1%, 34.5% and 27.6%, respectively); and the combined use of *ELMO1* and *ZNF582*, and the combination of three methylation markers showed the same sensitivity of 69.0% (Figure 3B). In EC patients, *ZNF582* also showed the highest sensitivity (72.9%) among those in *ELMO1* (27.1%) and *TFPI2* (39.6%). However, unlike the trend for *EMLO1* and *TFPI2* in GC and EJC, the sensitivity of *TFPI2* was about 1.5 times higher than that for *ELMO1* (Figure 3C). For all the UGC subjects, the sensitivities of each methylation markers showed a similar trend in GC, EJC and EC, and the combination of three methylation marker showed the highest sensitivity of 71.0%. As for specificity and Youden index, even though the specificity of *ELMO1* was as high as 100% (Figure 3E), its sensitivity was only 33.9%, resulting in a Youden index of 32.3% (Figure 3F). When combining the three methylation markers together, it showed the lowest specificity of 90.0% (Figure 3E), while it induced the best balance between sensitivity and specificity, with a Youden index of 61.0% (Figure 3F).

The AUC values for *ELMO1*, *ZNF582*, *TFPI2* and their combination for discrimination between the GC and control subjects were 0.638 (95% CI: 0.580–0.692), 0.843 (95% CI: 0.797–0.883), 0.628 (95% CI: 0.571–0.683) and 0.839 (95% CI: 0.792–0.879), respectively (Figure 4A). The AUC values of *ELMO1*, *ZNF582*, *TFPI2* and the methylation panel for the detection of EJC were 0.660 (95% CI: 0.594–0.723), 0.878 (95% CI: 0.827–0.918), 0.646 (95% CI: 0.579–0.709) and 0.837 (95% CI: 0.781–0.883), respectively (Figure 4B). For the detection of EC, the AUC values of *ELMO1*, *ZNF582*, *TFPI2* and the methylation panel were 0.603 (95% CI: 0.538–0.666), 0.889 (95% CI: 0.842–0.926), 0.689 (95% CI: 0.625–0.747) and 0.893 (95% CI: 0.847–0.930), respectively (Figure 4C). As for detection of the whole UGC group, the AUC values of *ELMO1*, *ZNF582*, *TFPI2* and the methylation panel were 0.632 (95% CI: 0.581–0.681), 0.861 (95% CI: 0.822–0.894), 0.647 (95% CI: 0.596–0.695) and 0.870 (95% CI: 0.832–0.902), respectively (Figure 4D). Furthermore, the sensitivities of DNA methylation panel for detection of UGC between different gender and age were shown in Table 2, and no significant difference was observed in the whole UGC group. However, a significant difference in sensitivities between different genders (*p* < 0.05) in the EJC samples was found, which might be due to the sample size of EJC being too small.

## 4. Discussion

UGC, as one of the most common cancer types in China, contributed about 16.0% (649,000) new cancer cases and 20.0% (482,400) new cancer deaths in China’s latest annual report [21]. Early detection of UGC can provide opportunities to implement strategies for effective treatment and improve the five-year survival rate. However, currently there is no clinically viable non-invasive method for the early detection of UGC. In this study, we selected three DNA methylation markers correlated to UGC and developed a plasma-based panel for UGC early detection.

*ELMO1*, encoding a member of the *ELMO* domain-containing protein family, plays an important role in promoting cell phagocytosis, reshaping and cell migration [22]. *ELMO1* has also been found to be associated with cancer development by regulating cancer cell proliferation, chemotaxis and invasiveness [22]. Therefore, *EMLO1* can be a potential diagnostic or prognostic biomarker for several cancer types [22]. Masahiro et al. found the methylation level of *ELMO1* in gastric cancer tissues was significantly higher than that in gastric atrophy tissues [23]. Bradley W. et al. indicated that the methylated *ELMO1* combined with other two methylation markers could detect 86% GC with a specificity of 95% in plasma [18]. Qin et al. also published a study involving the methylated *ELMO1* in EC patients, and the results showed methylated *ELMO1* could significantly distinguish ECs and controls in tissues and plasmas [24].

*ZNF582* is involved in the DNA damage response, cell apoptosis, differentiation and cell cycle control [25]. Recent studies revealed that methylated *ZNF582* in the promoter was an essential epigenetic mechanism for cancer silence [26], and it has been demonstrated that hypermethylation occurred in EC tissues [20,27]; however, the study of methylated *ZNF582* in EC plasma has not been reported. *TFPI2* belongs to the Kunitz-type serine proteinase inhibitor family, and methylation of *TFPI2* was found to be closely related to elevated cancer growth, invasion and dissemination [28]. Previous studies have demonstrated that *TFPI2* was frequently methylated in EC tissues [29] and GC tissues [30] or serums [31]; however, the application of plasma-methylated *TFPI2* for detecting UGC has never been reported.

UGC includes EC, EJC and GC. Although the methylation levels of *ELMO1*, *ZNF582* and *TFPI2* all showed a significant difference between each cancer type and control subjects, they still showed a preference for a certain cancer type. For example, *EMLO1* is more sensitive for GC, and *ZNF582* and *TFPI2* are more sensitive for EC. Combining three markers in a panel could narrow the sensitivity differences between the three cancer types and improve the sensitivities for detecting each cancer type. The sensitivities of methylated *ELMO1*, *ZNF582* and *TFPI2* for detecting UGC were 32.3%, 61.3% and 30.6%, respectively; and the combination for UGC detection in plasma yielded an increase of 38.7%, 9.7% and 40.3% in sensitivities compared with each one of the methylated markers.

In China, there are several challenges remaining in the current practice of the early detection of UGC, such as the low compliance of endoscopy and financial burden of the government. Although numerous non-invasive methods based on liquid biopsy biomarkers have been developed, most of these methods only targeted one cancer type [32,33]. As we all know, the cost of the current liquid biopsy method is still significantly higher than that of traditional serum tumor markers [34]; thus, it is difficult to apply those new methods in clinical practice in developing countries such as China. In this study, we detected three UGC types together for the first time and reacted the markers simultaneously in a single tube. Therefore, the time cost and the labor costs in our panel remain the same when compared with a single UGC type, such as GC or EC, while the whole cost for a single UGC type was reduced to 1/3. Moreover, the panel developed in our study also displayed a comparable sensitivity for each UGC type. For example, Xu et al. reported a combined methylation assay for GC detection with a sensitivity of 60.3% and a specificity less than 88% [35].

This study had some limitations. First, the sample sizes of EJC and EC were small compared with GC patients; thus, the overall performance might be affected by the unbalanced distribution of cancer types. Second, the age distribution among different groups may be biased due to the limited sample enrolment and did not reflect the true distribution in the real world. Third, the sensitivities of UGC across stages were not provided due to the stage information of some samples being unsuccessfully collected. Finally, control subjects should include a more high-risk interference sample, such as a Barrett esophagus or OLGA/OLGIM III-IV. Therefore, multi-center trials with a larger number of patient enrolment with complete information, as well as a prospective study within a large real-world population screening project, should be carried out in the future.

## 5. Conclusions

In conclusion, the results in this study indicated that the combination of several methylated markers could detect UGC with relatively high sensitivity and specificity, which might serve as a novel and potential alternative strategy for current UGC early detection. 

## Figures and Tables

**Figure 1 cancers-14-05282-f001:**
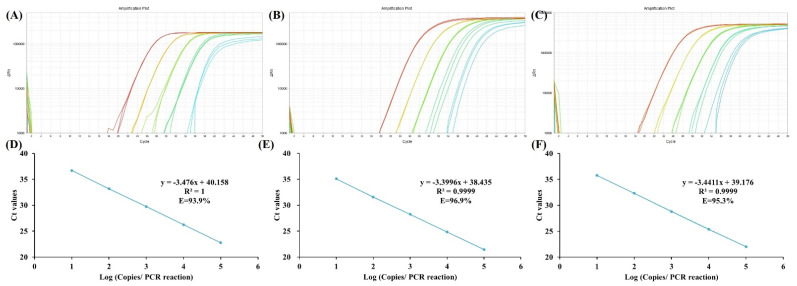
Amplification curve and PCR amplification efficiency by real-time qMSP of *ELMO1* (**A**,**D**), *ZNF582* (**B**,**E**) and *TFPI2* (**C**,**F**).

**Figure 2 cancers-14-05282-f002:**
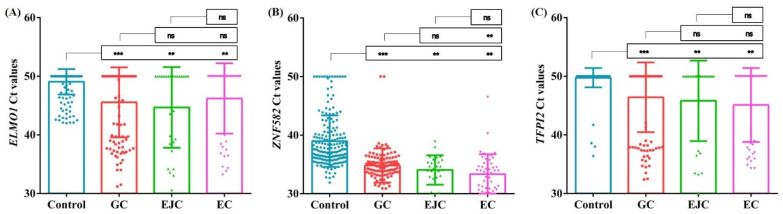
The DNA methylation levels of *ELMO1* (**A**), *ZNF582* (**B**) and *TFPI2* (**C**) in control, GC, EJC and EC plasmas. **, *p* < 0.01; ***, *p* < 0.0001, ns, no significant difference according to Student’s *t*-test.

**Figure 3 cancers-14-05282-f003:**
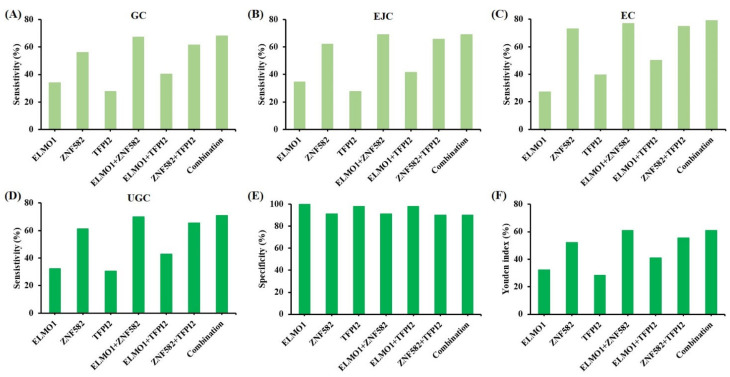
The sensitivities (**A**–**D**), specificity (**E**) and Youden index (**F**) of methylation panel in different samples.

**Figure 4 cancers-14-05282-f004:**
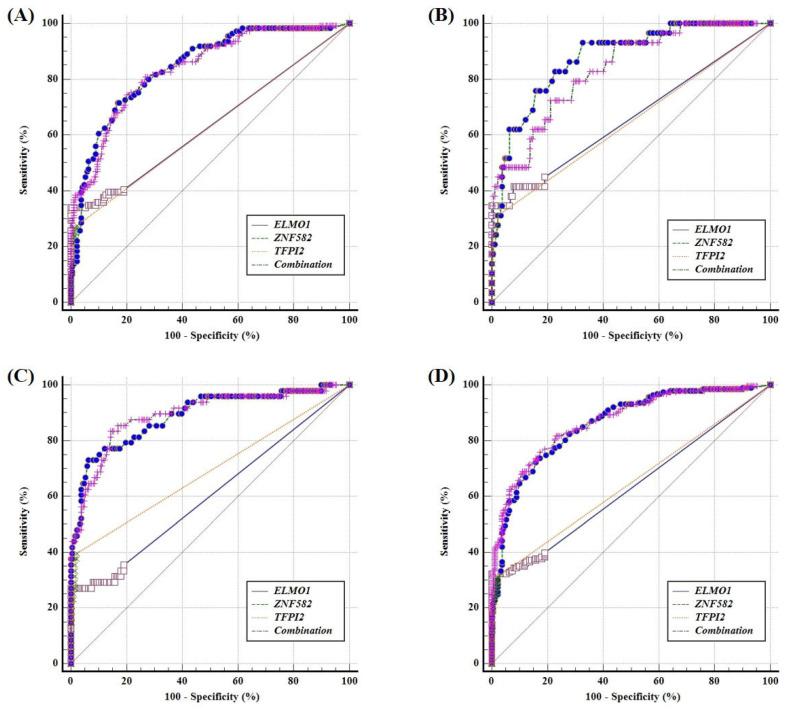
The ROC curves of methylation panel in different samples. (**A**) GC, (**B**) EJC, (**C**) EC, (**D**) UGC.

**Table 1 cancers-14-05282-t001:** Characteristics of participants enrolled in this study.

	Total Number	Gender		Age	
		Male (n, %)	Female (n, %)	Min–Max	Median
UGC	186	136 (73.1)	50 (26.9)	28–90	68
EC	48	42 (87.5)	6 (12.5)	52–90	68
EJC	29	22 (75.9)	7 (24.1)	50–87	71
GC	109	72 (66.1)	37 (33.9)	28–86	64
Control	190	74 (38.9)	116 (61.1)	23–79	45

**Table 2 cancers-14-05282-t002:** Sensitivities of methylation panel in different groups.

Characteristics		Total Number	Positive Number	Sensitivity (%)	*p*-Value
UGC					
Gender	Male	136	92	67.6	0.100
	Female	50	40	80.0	
Age	<60	47	31	66.0	0.629
	60–70	77	57	74.0	
	>70	62	45	72.6	
GC					
Gender	Male	72	47	65.3	0.415
	Female	37	27	73.0	
Age	<60	39	24	61.5	0.376
	60–70	41	31	75.6	
	>70	29	19	65.5	
EJC					
Gender	Male	22	13	59.1	0.042
	Female	7	7	100.0	
Age	<60	2	2	100.0	0.050
	60–70	11	5	45.5	
	>70	16	14	87.5	
EC					
Gender	Male	42	32	76.2	0.179
	Female	6	6	100.0	
Age	<60	6	5	83.3	0.622
	60–70	25	21	84.0	
	>70	17	12	70.6	

## Data Availability

The datasets used and/or analyzed for the current study are available from the corresponding author upon reasonable request.

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
