# Peer review of "A Novel Plasma-Based Methylation Panel for Upper Gastrointestinal Cancer Early Detection"

_cancers, 2022, doi:10.3390/cancers14215282_

Round 1
Reviewer 1 Report
Dear authors and editor,
The manuscript titled ‘’ A novel plasma-based methylation panel for upper gastrointestinal cancer early detections” try to develop cost-effective, non-invasive methods for early detection of upper gastrointestinal cancer. The DNA methylation levels of ELMO1, ZNF582 and TFPI2 were analyzed in upper gastrointestinal cancer from different regions(GC, EJC and EC). The topic of the project is very interesting and extremely important from a clinical point of view. Importantly, the authors see the limitations of the project and results. In my opinion, the three above-mentioned cancers of the upper gastrointestinal tract usually have different molecular backgrounds and therefore should be considered / analysed separately. As the clinic shows, the method of comprehensive treatment is also slightly different.
The paper is well-organised, the language is correct and the content is understandable. Literature properly selected and up to date. Most them are from the last 10 years
However the manuscript is good, I have some comments that should be clarified:
- It seems that the authors do not fully understand what a retrospective project means.
- I have not found the consent number of the bioethics committee.
-Before performing experiments, there is a lack of exact information on why the authors chose only ELMO1, ZNF582 and TFPI2 methylation regions. In my opinion, this is the basis of the entire project, therefore must be done.
- Too few cases to draw general conclusions. For this reason, the statistics are distorted.
In conclusion, manuscript requires a thorough improvement. The project is extremely interesting! The results presented in the manuscript should be treated as preliminary research.
Thank you for your choice me as a reviewer.
Author Response
Please see the attachment, thank you.

Reviewer 2 Report
Dear Editor
The manuscript is well written and non invasive biomarkers are highly relevant for diagnosis of esophageal and gastric cancer. The DNA methylation is one of the most interesting ways to detect upper gastrointestinal cancers before upper endoscopy.
Comments:
1. Introduction: Please include other non invasive biomarkers described in the literature (e.g. miRNA) and the advantage using DNA mathylation biomarkers for the diagnosis.
2. If there is a difference with this biomarkers for early vs advanced stages of GI cancers. If there is evidence of DNA methylation (e.g. Reprimo-like as a marker of hypermethilation of the DNA) in terms of prognosis (as a marker of survival, or response to specific chemotherapy or immunotherapy.
3. Methods:
Please describe percentage of early/advanced GI cancers
Please describe if control also had upper endoscopy. If they have upper endoscopy, please include information related to premalignant lesions (e.g. Barrett esophagus; gastric intestinal metaplasia) because premalignant lesions could have abnormal levels of DNA methylation.
Results:
Please include subanalysis for Dysplasia/early vs advanced stages of cases and (low risk controls: histological/endoscopically normal vs high-risk controls: with premalignant lesions such as Barrett esophagus or OLGA/OLGIM III-IV)
Author Response
Please see the attachment, thank you.

Round 2
Reviewer 1 Report
Dear authors and editor,
The corrected manuscript is improved.
The authors have included all my correction/suggestion in the presented version of the manuscript and They comprehensively answered the questions posed in the review.
In conclusion, I support publication of the presented article.
Thank you for your choice me as a reviewer.